# Biodegradable Nanoparticles Loaded with Levodopa and Curcumin for Treatment of Parkinson’s Disease

**DOI:** 10.3390/molecules27092811

**Published:** 2022-04-28

**Authors:** Bassam Felipe Mogharbel, Marco André Cardoso, Ana Carolina Irioda, Priscila Elias Ferreira Stricker, Robson Camilotti Slompo, Julia Maurer Appel, Nathalia Barth de Oliveira, Maiara Carolina Perussolo, Claudia Sayuri Saçaki, Nadia Nascimento da Rosa, Dilcele Silva Moreira Dziedzic, Christophe Travelet, Sami Halila, Redouane Borsali, Katherine Athayde Teixeira de Carvalho

**Affiliations:** 1Advanced Therapy and Cellular Biotechnology in Regenerative Medicine Department, The Pelé Pequeno Príncipe Research Institute, Child and Adolescent Health Research & Pequeno Príncipe Faculties, Curitiba 80240-020, Brazil; bassamfm@gmail.com (B.F.M.); marcoacardoso@yahoo.com.br (M.A.C.); anairioda@gmail.com (A.C.I.); priscilaeferreira@gmail.com (P.E.F.S.); robsoncamilotti@gmail.com (R.C.S.); juliamappel@gmail.com (J.M.A.); nathybarth03@gmail.com (N.B.d.O.); perussolo10@gmail.com (M.C.P.); claudiasacaki@gmail.com (C.S.S.); nadianr@gmail.com (N.N.d.R.); dilceledz@gmail.com (D.S.M.D.); 2Centre de Recherches sur les Macromolécules Végétales (CERMAV), Centre National de la Recherche Scientifique (CNRS), Université Grenoble Alpes, F-38000 Grenoble, France; christophe.travelet@cermav.cnrs.fr (C.T.); sami.halila@cermav.cnrs.fr (S.H.); redouane.borsali@cermav.cnrs.fr (R.B.)

**Keywords:** nanoparticles, glutathione, Parkinson’s disease, L-DOPA, curcumin

## Abstract

**Background:** Parkinson’s disease (PD) is the second most common age-related neurodegenerative disorder. Levodopa (L-DOPA) remains the gold-standard drug available for treating PD. Curcumin has many pharmacological activities, including antioxidant, anti-inflammatory, antimicrobial, anti-amyloid, and antitumor properties. Copolymers composed of Poly (ethylene oxide) (PEO) and biodegradable polyesters such as Poly (ε-caprolactone) (PCL) can self-assemble into nanoparticles (NPs). This study describes the development of NH_2_–PEO–PCL diblock copolymer positively charged and modified by adding glutathione (GSH) on the outer surface, resulting in a synergistic delivery of L-DOPA curcumin that would be able to pass the blood–brain barrier. **Methods:** The NH_2_–PEO–PCL NPs suspensions were prepared by using a nanoprecipitation and solvent displacement method and coated with GSH. NPs were submitted to characterization assays. In order to ensure the bioavailability, Vero and PC12 cells were treated with various concentrations of the loaded and unloaded NPs to observe cytotoxicity. **Results:** NPs have successfully loaded L-DOPA and curcumin and were stable after freeze-drying, indicating advancing into in vitro toxicity testing. Vero and PC12 cells that were treated up to 72 h with various concentrations of L-DOPA and curcumin-loaded NP maintained high viability percentage, indicating that the NPs are biocompatible. **Conclusions:** NPs consisting of NH_2_–PEO–PCL were characterized as potential formulations for brain delivery of L-DOPA and curcumin. The results also indicate that the developed biodegradable nanomicelles that were blood compatible presented low cytotoxicity.

## 1. Introduction

Parkinson’s disease (PD) is the second most common age-related neurodegenerative disorder and represents a growing healthcare concern with elderly populations. The disease is associated with a range of symptoms, including bradykinesia, rigidity, tremor, dementia, and depression [1]. Levodopa (L-DOPA) is the gold-standard drug for PD treatment, and its use results in a marked improvement in patient quality of life, but just for a limited period. L-DOPA shows tolerance and the development of induced dyskinesias during treatment. Fluctuations of L-DOPA blood levels are related to the intermittent stimulation of dopamine receptors, resulting in a discontinuous response with “On” and “Off” periods of action (wearing off) and disease progression [2]. Moreover, the metabolism of L-DOPA also generates a variety of free radicals that increase the loss of nigrostriatal dopaminergic neurons and the development of the disorder [3,4].

The blood–brain barrier (BBB) limits access to therapeutic molecules and macrostructures [5]. Brain-targeted drug delivery is a significant concern since the BBB permeability is crucial for nanoparticles (NPs) to exhibit a therapeutic effect. Thus, one great therapeutic challenge in PD treatment is developing an effective drug targeted system capable of improving the symptoms, extending the brain delivery of L-DOPA by crossing the BBB, avoiding fluctuations in its concentration, and reducing the rate of neurodegeneration.

Nanosized polymeric micelles formed by amphiphilic copolymers with A (hydrophilic)-B (hydrophobic) diblock structures could be employed as vehicles for drug administration to the brain once they can be designed to display different properties of targeting, pharmacokinetics, and cargo release of drugs, including insoluble or poorly soluble compounds and surmount the BBB [6].

Amphiphilic block copolymers composed of poly(ethylene oxide) (PEO) and biodegradable polyesters such as Poly(ε-caprolactone) (PCL), which can self-assemble into nanoparticles (NP) in aqueous medium, have gained much attention in the nanomedicine field. PEO is an FDA-approved biodegradable polymer and a common constituent for the hydrophilic outer shell of nanoparticles. It possesses a great number of useful physicochemical and biological properties, including hydrophilicity, solubility in water, lack of toxicity, and absence of antigenicity and immunogenicity [7]. PCL is one of the most widely used FDA-approved biodegradable polymers because of its biocompatibility, biodegradability, mechanical properties, non-toxicity, high drug permeability, and slow in vivo degradation properties. PCL has a wide spectrum of applications in the biomedical field, including formulations for drug delivery as nanocapsules capable of prolonging the drug release and enhancing the drug stability [8].

Therefore, to modify and improve pharmacological and therapeutic effects, drugs can be encapsulated in PEO/PCL-based NPs based in noncovalent interactions as hydrogen bonding and hydrophobic or ionic interactions and released by diffusion directly at a specific tissue or site of action [7]. Furthermore, NPs have also demonstrated the ability to deliver antioxidant compounds that reduce oxidative stress in various diseases, including PD [9,10].

Curcumin (CUR) is a natural low-molecular-weight hydrophobic polyphenolic phytoconstituent that is isolated from the perennial herb *Curcuma longa*, with various pharmacological properties [11,12]. The development of nanotechnology-based delivery systems of CUR demonstrated its neuroprotective effect in Parkinson’s disease models [13,14].

The approaches for active targeting of NP to overcome the BBB involve different mechanism for the transport of macromolecules across the BBB, as the absorptive-mediated transcytosis (which comprises the use of positively charged moieties), the transporter-mediated transcytosis (correlated with nutrients or substrates, such as glutathione and glucose, among others), and the receptor-mediated transcytosis (associated with the facilitated passage of targeting ligands). Therefore, endocytic pathways can be activated by using direct moieties or ligands as vectors [15,16].

Glutathione (GSH) is a water-soluble endogenous tripeptide of glutamic acid, cysteine, and glycine that possesses antioxidant-like properties with an active uptake transporter highly expressed at the BBB. Once specific binding sites of BBB receptors for GSH are selective compared with other endogenous peptides, GSH can be used as a safe, effective, and specific ligand that can target and enhance drug delivery of NPs to the brain without toxicity. However, the detailed mechanisms of GSH as a ligand mediating endocytosis need to be elucidated [17,18].

More recent studies have also demonstrated that the use of GSH as a targeting ligand to deliver NPs inside the brain has enhanced their neuronal bioavailability and therapeutic effects, improving treatment outcomes and cellular internalization of different nanoformulations, including nanomedications undergoing clinical evaluation such as 2B3-101 [19,20,21,22,23].

This manuscript describes the development of NPs composed of NH_2_–PEO–PCL diblock copolymer positively charged and modified by the addition of GSH on the outer surface, obtaining a dual functionalized system for a synergistic delivery of L-DOPA and CUR. This dual system would be able to pass the BBB, target the brain tissue, and provide a more sustained release of drugs for potential application in PD treatment.

## 2. Results and Discussion

### 2.1. Characterization of NH_2_–PEO–PCL Nanoparticles

The NTA analysis, a technique that enables the visualization, sizing, and quantification of nanoparticles in suspension by using a highly sensitive video camera, demonstrated a narrow distribution of the NH_2_–PEO–PCL NP and a size of 99.5 + 7.3 nm, smaller than those obtained by DLS (Figure 1).

The results show that the sizes of the reported NH_2_–PEO–PCL NPs were bigger than those reported by Numata (2015) [24]. The NPs’ formation during nanoprecipitation is primarily governed by the diffusion stranding phenomenon and the Marangoni effect. Thus, different preparation conditions and parameters on PCL NP preparation by solvent displacement can affect the nanoparticle size, as described by Badri et al. (2017) and Mora-Huertas et al. (2011) [25,26]. The physicochemical properties of solvents used in NP production, such as viscosity and water miscibility, are important factors that control the size and size distribution of the NP generated. Once THF is more viscous than acetone, the former can retard the diffusion of solute and solvent molecules during the mixing procedure of solvent displacement method and, thus, generate larger particles, in a similar way described by Tam et al. (2016) [27].

A further analysis of the DLS and SLS scattering data was performed. The shape factor, ρ, was obtained from the ratio ρ = Dg/Dh. The ρ values of 1.4 for NH_2_–PEO–PCL NP fall between those of vesicles (ρ = 1) and rigid rods (ρ ≥ 2.0), meaning that the overall shape of these NPs was spherical. According to Gross et al. (2016) [28], the shift in size could be explained because the size distributions obtained by DLS consist of weight distributions, whereas those obtained by NTA are number distributions. 

The preparation method of nanoparticle aqueous suspensions formed by amphiphilic block copolymers, called nanoprecipitation or the solvent displacement technique, is the dissolution of the copolymer in a mixture of good solvents for both blocks, followed by the decrease of the organic solvent content by the injection of the solution in an excess of water. A further decrease in the content of the organic solvent, by evaporation or by dialysis of the solution against water, causes the collapse of hydrophobic cores, generating stable NPs [29].

The nanoprecipitation method described by Y. Numata et al. (2015) [24], with small modifications, was used for the preparation of NH_2_–PEO–PCL NPs. In order to determine the minimization of the scattering intensity corresponding to the most dispersed polymer state in solution, the effect of the different organic solvent percentages (from 50% to 100% of THF in water) was studied in the polymer dissolution. The formation of NH_2_–PEO–PCL aggregates formed in THF/water solutions decreased with the increasing percentage of THF, reaching a minimum at 90% of THF. Transmission electron microscopy determines the size of dry particles, while DLS determines the hydrodynamic diameter of particles in water. Because amphiphilic block polymeric micelles always have a loose structure in water, the particle size determined by DLS is always slightly more extensive than that determined by transmission electron microscopy [30]. TEM observed that the NH_2_–PEO–PCL NP was the systems’ spherical shape, as shown in Figure 2. Nanoparticles with a size less than 200 nm can be injected intravascularly, permeate and traffic through different tissues, bind to cell surface receptors, and enter to target cells for intracellular drug delivery [31]. This result demonstrates that the NH_2_–PEO–PCL NPs are larger than the threshold for glomerular filtration in the kidney (Mw approximately 50 kDa) and smaller than 200 nm, increasing the blood circulation and the tissue targeting effect, especially to the brain [32].

PDI is an indicative of the NPs size distribution, ranging from 0 to 1, and defines the dispersion homogeneity. Values close to 1 indicate heterogeneity and PDI values lower than 0.5 indicate NPs of better quality with a homogenous distribution of particle size [33,34]. All the NPs showed unimodal distribution, as indicated with PDI values of less than 0.35, suggesting a homogenous distribution under the preparation conditions.

Zeta potential (ZP) analysis can provide very useful information on the surface properties and stability of the NP system.

The ZP of the NH_2_–PEO–PCL NPs was +25.6 + 0.5 mV (Table 1). This positive value is attributed to the formation of quaternary ammonium groups on NH_2_–PEO chains generated by its protonation in aqueous medium and, as expected, suggest the orientation of the PEO chains to the outside of the micellar NPs. This absolute value of the ZP also implies a high stability of the NPs suspension once the repulsive forces generated by the positive charges surrounding the NPs prevent its aggregation and the chances of coalescence of NPs in the system [35].

In addition, the ZP of NPs is an essential factor in determining the in vivo interactions of nanoparticles with the cell membrane. The positive charge of NP complexes can help the NPs bind tightly to the negatively charged cellular membrane, having a higher tendency to attach and internalize into the cells by endocytosis, compared to negatively or neutrally charged particles [36].

In order to evaluate the comportment of the ZP of the NH_2_–PEO–PCL NPs in different ionic strengths, the NPs were submitted to a crescent NaCl concentration ranging from 1 mM to 2 M (Figure 3). WE observed a progressive drift toward neutral, or slightly negative, values for ZP that progressively neutralized the positive charges of NH_2_–PEO–PCL NPs and confirmed its positive value.

This implies that the presence of NaCl in the aqueous medium substantially influenced the properties of the NPs that were measurable by ZP and can be due to the electrostatic interactions after salt addition with the surface of the NPs, resulting in lower values of electrophoretic mobility [37].

The stability of NH_2_–PEO–PCL NPs in diverse NaCl concentrations of the medium was also determined by measuring of the particle size by DLS. As depicted in Figure 3, the NH_2_–PEO–PCL NPs were stable in a salt concentration superior to the physiological salt conditions (150 mM NaCl), showing just a slight variation in the size throughout all the salt concentrations evaluated. These results demonstrate the maintenance of NP suspensions in an aqueous medium, due to the steric repulsive forces of the NH_2_–PEO corona and the hydrophobic interactions from the core of PCL, resulting in a stable system.

### 2.2. Curcumin (CUR) Encapsulation

A series of curcumin-loaded NPs formulated with different amounts of CUR (2, 3, 4, and 5 mg), added during the NPs preparation, were made in order to evaluate the CUR encapsulation efficiency. When the CUR-loaded micelles were prepared with 5 mg of CUR, the systems became instable, aggregating in a few minutes. On the other hand, at the drug-to-copolymer ratio ranging from 0.2 to 0.4 (2, 3, and 4 mg of CUR), no precipitate was detected, and the successful encapsulation was observed by the formation of a clear yellow solution of CUR loaded into micelles, as observed by Shao et al. (2011) [38], Wang et al. (2013) [30], and Scarano (2015) [39].

As described by Chow et al. (2015) [40], if the concentration of copolymer is low, relative to CUR in the organic phase, part of the hydrophobic CUR will be exposed to the external aqueous medium, due to an insufficient surface coverage of the NPs by the copolymer, resulting in particle aggregation, thus compromising the stability [40]. Moreover, the nanosuspension system prepared with a lower drug-to-copolymer ratio becomes more stable when the CUR-to-copolymer ratio decreases, and a more considerable amount of the copolymer covers the surfaces of the NP, increasing its steric stabilization [27]. After these results, the lowest tested CUR-to-copolymer ratio (0.2) was taken as the optimized formulation for the NP suspension preparation.

The drug loading and encapsulation efficiency of curcumin-loaded NPs, prepared with 2 mg of CUR, were 98.3% ± 0.9% and 19.8% ± 0.2%, respectively, in accordance with the results described by Gong et al. (2013) [11], Wang et al. (2013) [30], Mazzarino et al. (2014) [41], and Mogharbel et al. (2018) [42].

After the encapsulation, the size of the NPs increased as the CUR-to-copolymer ratio increased (129.3 + 6.5, 148.5 + 6.8, and 149.6 + 8.9 nm, to a copolymer/CUR ratio of 5, 3.3, and 2.5, respectively). This result was expected, because the incorporation of CUR into the hydrophobic cores increased the volume of the NPs [11,41,43], resulting in micelles with swollen cores forming spontaneously. The monodispersity was confirmed by polydispersity indices lower than 0.3 in all samples.

The CUR-loaded NP suspension was transparent and completely dispersed in aqueous media, with no aggregates, indicating full dispersibility of curcumin, while free CUR exhibited poor aqueous solubility, as described by Wang et al. (2013) [30]. Therefore, the encapsulation of CUR into polymeric micelles resulted in a homogenous and stable dosage form in aqueous solution with high drug loading and small particle size, making CUR administration possible.

As suggested by the results obtained by the determination of ZP, amphiphilic NH_2_–PEO–PCL NPs were formed by a core–shell structure in water, constituted by the hydrophobic PCL chains in the core and the hydrophilic PEG chains oriented to the shell. Consequently, hydrophobic and water-insoluble drugs such as CUR can be encapsulated into the hydrophobic core by the hydrophobic interactions with PCL, and because of the surface hydrophilic shell of PEG, and became more useful in biological systems [43,44].

### 2.3. L-DOPA Encapsulation

The drug loading and encapsulation efficiency of L-DOPA-loaded NPs were 12% ± 1.4% and 3.6% ± 0.4%, respectively, which was reasonable considering the hydrophilic nature of dopamine. These values correlated with the results obtained by Arica et al. (2005) [45], with drug loading values ranging from 14% ± 1.2% to 20% ± 1.4%. Shin et al. (2014) [46] also published similar results for the drug loading of dopamine, the neurotransmitter originating from the decarboxylation of L-DOPA, with values ranging from 2.7% ± 1.8% to 18.6% ± 9.2%. No significant differences were observed for the size of unloaded and L-DOPA-loaded nanoparticles.

The advantages of PCL for drug release applications include its high permeability to drugs, slow and sustained release of entrapped therapeutic compounds, less acidic degradation products as compared to other types of aliphatic polyesters, and absence of systemic toxicity [8,47].

The results of L-DOPA encapsulation have demonstrated that NH_2_–PEO–PCL NPs can encapsulate L-DOPA and, due to the PCL physicochemical characteristics, are also suitable for long-term sustained delivery of this bioactive agent. Consequently, it is expected that the developed NPs would be able to reduce the toxicity associated with L-DOPA; preserve the structural integrity of the encapsulated drug; enhance the drug stability, protecting it from rapid peripheral metabolism; and reduce the side effects associated with the L-DOPA treatment.

When loading both L-DOPA and curcumin (LdCurNP) in the same micelle, the drug loading and the encapsulation efficiency for L-DOPA were 10.4 ± 1.5% and 3.1 ± 0.5%; and for curcumin, they were 97.7 ± 1.0% and 19.5 ± 0.2%, respectively (Table 2).

### 2.4. GSH Coating of the NPs

Glutathione (GSH) coating promoted a slightly increase in the average size of the NH_2_–PEO–PCL NPs, from 117.3 + 4.6 to 128.6 + 1.2. This is in agreement with the results reported previously by Mdzinarishvili et al. (2013) [48] and Geldenhuys (2011) [49], who used the same method for the functionalization of the NPs, as for the NPs produced by using chitosan conjugated with GSH for oral drug delivery proposed by Chronopoulou, et al. (2016) [50].

TEM observations of the GSH-coated NH_2_–PEO–PCL NPs demonstrated that, after its functionalization, the system maintains the spherical shape, as shown in Figure 2.

The NTA analysis of the GSH-coated NH_2_–PEO–PCL NPs demonstrated a size of 99.5 + 7.3 nm, smaller than those obtained by DLS (Figure 4).

This surface functionalization also gave rise to a significant decrease in the Zeta potential of the NP, from +25.6 + 0.5 to +11.0 + 0.4 mV, suggesting an effective masking effect on the exposed positively charged NH2- groups of PEO. This result was attributed to the formation of a coating of GSH adsorbed to the positively charged surface of NP, in a similar way described by Duxfield et al. (2016) [51].

The surface modification in our system was achieved by physical adsorption of GSH to the outside surface of NP; this is a simple approach that provides stability to the NP suspensions and sustains their bioactivity, due to ionic interaction between the positive charges of quaternary ammonium groups on NH_2_–PEO chains outside of NP and the negative charge of GSH and reinforced by the hydrogen bonds between ether, hydroxyl, and amino groups from PEO and GSH.

The GSH-adsorbed NPs were dialyzed against ultrapure water for two days by changing the water four times each day to evaluate the NPs’ stability after their coating. It was observed that, after the dialysis, the ZP of the NPs remained unchanged (+11.2 + 1.6 mV), indicating that the interaction between the NPs surface and GSH is strong, making the functionalized system stable. This reveals that the GSH-coated NPs still retained a desirable positive charge at their surface.

No significant differences were observed for the unloaded and loaded GSH coated NP ZP values. GSH was coated onto the curcumin and L-DOPA NPs after their formation by the nanoprecipitation method coating the particles with GSH that would not modify the drug encapsulation.

Taking all of these results together, we hypothesized that the GSH-coated loaded NPs would be favorable for their transportation across the BBB by two different biological mechanisms.

First, the adhesion and transport properties of the NPs can be facilitated by the electrostatic attractions between the positively charged samples and the negatively charged cell membranes by adsorptive-mediated transcytosis (AMT). AMT is a vesicular transport route of cationic substances through the BBB, and, in contrast to receptor-mediated transcytosis, it does not require specific binding sites on cell surfaces but involves electrostatic interactions between polycationic substances and negative charges on the endothelial surface [52,53].

Second, GSH coating at loaded NPs can utilize the existing GSH transport mechanism in the BBB, contributing to the increase the permeability of positive charged NPs into the brain. GSH acts as an endogenous ligand for the NMDA and AMPA receptors present in the BBB, enabling the coated NPs to bind the NMDA receptor at glycine and glutamate site, enhancing the clathrin-mediated endocytosis-coated NP–NMDA receptor complex inside the cerebral cells.

Moreover, GSH glutathione-conjugated NPs could be recognized as endogenous units decreasing the clearance out of the brain and allowing the NPs to remain inside the brain for a longer time, enhancing its therapeutic effect [18,54]. In addition, authors have successfully demonstrated the delivery of GSH-conjugated NPs inside the brain as a safe, effective, and specific ligand for brain-targeted drug delivery systems that are capable of improving the transport of drugs across the blood–brain barrier [17,53,55].

### 2.5. Freeze-Drying Stability

After lyophilization, the observation of NH_2_–PEO–PCL NPs revealed the formation of an intact cake occupying the same volume as the original frozen volume only in the samples which used hydroxypropyl-β-cyclodextrin (HPbCD) as a cryoprotectant.

The DLS analysis of the NPs submitted to lyophilization confirmed that, among all tested cryoprotectants, only HPbCD was capable of preserving the NPs during lyophilization. For the other protectants, the redispersed suspensions showed different extents of particle size and some visible precipitates.

The superior protectant performance of HPbCD relative to the other four protectants could be linked to its inherently non-crystalline nature and better adsorption onto the NPs surface during the sublimation of the aqueous phase. In addition, HPbCD is a collapse temperature modifier with a relatively high Tg’ (i.e., the glass transition temperature of maximally cryo-concentrated solution), which renders it especially useful for raising the overall collapse temperature the nanoparticle formulation, as well as shortening the primary drying cycle. Moreover, HPbCD is an atoxic cyclic oligosaccharide used to improve the water-solubility and bioavailability of medicinal products currently found in marketed parenteral formulations [40].

One of the desired characteristics of the lyophilized NPs includes an intact cake occupying the same volume as the original frozen mass, which facilitates the rapid reconstitution of lyophilized product in a solvent system. A significant drop in volume may indicate formulation collapse, which is unacceptable from a gross macroscopic perspective. Apart from the lyophilization process, lyophilized cake formation is highly dependent on the composition of the formulation, including cryoprotectant [56]. Results obtained by other authors showed that HPbCD was also the most effective cryoprotectant among different sugars [40,57,58].

The reconstitution time of lyophilized NPs with HPbCD was practically instantaneous following the addition of water and gentle manual shaking by inversion. The NPs were readily dispersible, showing uniform distribution and no aggregation. The rehydration was achieved upon vortexing and sonication for the other collapsed formulations, which took a long reconstitution time.

The level of cryoprotection provided by sugars generally depends directly on their concentrations, meaning that, the higher the concentration of cryoprotectant, the better the stability of the nanoparticles. Moreover, it is essential to optimize the concentration of the used cryoprotectant, to use a minimum concentration while preserving all desirable characteristics [40].

As HPbCD was demonstrated to be the best protectant for the NPs during freeze-drying, eight different concentrations of HPbCD, ranging from 0.25 to 5% *w*/*v*, were tested. Precipitates were observed upon redispersion of the freeze-dried samples if the HPbCD concentration was below 2%. Then a minimum concentration of 2% of HPbCD was considered suitable for the reconstitution of lyophilized GSH-coated NH_2_–PEO–PCL NPs. All the samples tested with concentrations of HPbCD ranging from 2 to 5% did not showed signs of aggregation and were easily reconstituted by manual shaking; in addition, the size measurement demonstrated an almost-identical mean particle size before and after freeze-drying. This result demonstrated no further enhancement of particle stability with higher HPbCD concentrations, as Chow et al. (2015) [40] presented.

After optimizing the lyophilization process and selecting the cryoprotectant, the results indicated that HPbCD at a concentration of 2% *w*/*v* was superior among all other sugars studied, resulting in desirable lyophilized NH_2_–PEO–PCL NPs. The NPs’ Zeta potential values were not statistically significant, confirming that the stabilization and NPs’ outside charges were not affected.

### 2.6. Cytotoxicity Evaluations of the Loaded Nanoparticles

When evaluating the biocompatibility of newly designed nanoparticles, ensuring safety in vitro assays is necessary before advancing in preclinical/clinical trials [59]. This study performed three nanotoxicity assays: erythrocyte hemolysis, MTT, and the LIVE/DEAD^®^ viability assay. The concentrations, ranging from 1 to 100 µM) of the L-DOPA and curcumin-loaded NH_2_–PEO–PCL GSH-coated NPs (LdCurNP), applied in these studies were calculated based on the amount of loaded L-DOPA in nanoparticles, taking into account the encapsulation efficiency and drug loading previously described. Because of this difference, curcumin concentrations could be as high as five-fold the concentration of L-DOPA, and, thus, Appendix A are available showing the assays using only curcumin-loaded nanoparticles (CurNP) ranging from 200 to 500 µM (Appendix A).

#### 2.6.1. Erythrocyte Hemolysis Assay

In the present study, the centrifuged tubes demonstrated that none of the groups had similar hemolytic activity, as seen in the positive control sample (Figure 5A).

After calculating the hemolysis rate for each NP suspension concentration, it was detected that the hemolysis rate was lower than 5% for all concentration groups tested. Only a few concentrations were slightly over 2% in Sample 1 (Figure 5B and Table 3). In Sample 2, almost all the concentrations were above 2%, but none passed the threshold of 5% (Figure 5C and Table 3). Such results suggest that the unloaded and L-DOPA and curcumin-loaded NH_2_–PEO–PCL GSH-coated NPs have good blood compatibility.

Since all administration routes of NPs lead to blood circulation, blood compatibility is one of the most important parameters when developing a new nanoparticle [60,61]. The standard ASTM-F765 [62] defines hemolytic rates of 0 to 2% as non-hemolytic, 2 to 5% as slightly hemolytic, and above 5% as hemolytic.

As a complement for the absorbance data, it is also important to record images of the centrifuged tubes for qualitative evaluation of the solutions in order to avoid false negatives [63].

The development of co-polymer NPs can be made within numerous combinations of different structures; this makes the comparison between them much more complex. It should consider the types of polymers that were used, if the NPs were coated, and if they were loaded or not with drugs.

Regarding all the differences between the NPs, it is essential to encounter a common ground between them. In the present study, the hemolytic results were validated by hemolysis assays for polymeric nanoparticles from Mazzarino et al. (2015) [64], who developed NPs with xyloglucan-block-polycaprolactone and also obtained good blood compatibility, and Fan et al. (2018) [65], who evaluated a co-polymeric curcumin-loaded NPs synthetized with poly (lactic-co-glycolic acid) and polyethylene glycol and also obtained good blood compatibility.

#### 2.6.2. MTT Cytotoxicity Assay

The MTT assay is a widely used colorimetric experiment that evaluates the metabolic capacity of cells, more specifically, the capacity of oxidoreductases to split the tetrazolium ring into the purple-colored formazan [60,66,67].

In this assay, Vero and PC-12 cells were exposed to unloaded or L-DOPA and curcumin-loaded NH_2_–PEO–PCL GSH NPs suspensions in different concentrations. The Vero cell line is often applied in general cytotoxic evaluations [42,68]; on the other hand, the PC-12 cell line, when differentiated, is mainly used when neuronal cytotoxicity needs to be observed [69,70].

In our findings related to the Vero cell line, at 24 h of incubation, only the concentration 10 µM of the LdCurNPs had a significant decrease in absorbance compared with the control.

However, at 48 and 72 h, both unloaded and loaded nanoparticles suspensions induced a decrease in the proportional absorbance compared to the control group in most of the concentrations tested, except UnNP at 48 h with 1 and 100 µM (Figure 6A, 48 and 72 h). A significant absorbance decrease was observed with Vero cells when comparing the unloaded versus the loaded nanoparticles (comparisons represented by # in Figure 6A), and this could be related with the slower drug delivery mechanism of nanoparticles. This decrease is also present in the assay with only curcumin-loaded nanoparticles (Appendix A) and is corroborated with other authors that also applied the MTT assay and treating Vero cells with curcumin. Kong (2009) [71] observed a ratio decrease after 48 h of exposure of the Vero cells with curcumin at a concentration of 20 µM, and Prasetyaningrum (2018) [72] also reported a decrease in this same cell line, with around 30 µM of curcumin. Both studies used curcumin powder solubilized in DMSO, leading to an all-at-once exposure to these concentrations. When a drug is loaded into nanoparticles, the drug liberation is time-dependent, so it can be hypothesized that the curcumin liberation from the nanoparticles was sustained throughout the assay, maintaining a steady concentration.

Although there are reports in the literature that curcumin had no cytotoxicity effects in undifferentiated PC12 cells [73,74], some other studies have shown the opposite, [75,76,77]. Mendonça (2013) [76] also reported the cytotoxicity effects of curcumin in differentiated PC12 cells, but lower in undifferentiated cells.

Farani (2019) [78] tested a curcumin-loaded iron oxide G-NH_2_ nanocarrier bonded with PEG in undifferentiated PC12 cells, with concentrations of the curcumin up to 134 µM and 48 h of incubation, and their results showed a cell viability above 80% even in higher concentrations.

Most of these studies treated cells with curcumin dissolved in DMSO solution and evaluated only 24 h of incubation before analysis. In the present study, we wanted to verify if long-term exposure (48 and 72 h) of L-DOPA and curcumin-loaded nanoparticles resuspended directly into the cell culture medium could somehow affect cells’ viability.

When treating PC12 cells with L-DOPA (100 µM) for 24 h, studies have shown a decrease in viability when compared with the control group [79,80,81], thus corroborating our results.

It is also reasonable to consider that it is not entirely understood how suspended nanoparticles interact with cell membranes, organelles, and nuclei, and if such interactions can lead to cytotoxicity [82].

For the MTT assay, there is a significant difference when comparing the unloaded and the loaded nanoparticles in the Vero cell line; and this is accentuated in 48 and 72 h. These differences appear in fewer comparisons in the PC12 cell line (Figure 6B).

These considerations are important because there is a lack of works in the literature that report a longer exposure times of L-DOPA, and curcumin-loaded nanoparticles. Although the MTT analysis detected cytotoxicity, especially in longer exposure times (48 and 72 h), it is necessary to discuss that, besides being extensively used in cytotoxicity assays, this colorimetric assay has its flaws, such as over/underestimation of absorbance, susceptibility of microplate edge effect, particle-induced artifacts, and false positives [83,84,85,86,87].

#### 2.6.3. LIVE/DEAD Viability Assay

The LIVE/DEAD assay, together with high throughput microscopy, consisted of a more accurate representation of the cell population, regarding the viability, and can complement colorimetric assays. If cells are viable, they can convert, by intracellular esterases, the nonfluorescent calcein—AM into the green fluorescent calcein. Nevertheless, when cells are compromised and their cell membrane is damaged, the nuclei marker ethidium homodimer-1 binds with nuclei acids and emits a red fluorescence [67,88].

After analyzing the LIVE/DEAD viability assay of the Vero cell line treated with LdCurNP in different concentrations, it was observed that the viability was more than 95% in all concentrations and all time points. Although some concentrations had a significant difference compared with the control group, that cannot be biologically relevant (Figure 7B). An increase in cell number can also be observed between time points, suggesting that cells are viable and proliferating (Figure 7A).

Considering the analyzed LIVE/DEAD viability assay of PC12 cell line treated with LdCurNPs in different concentrations, it was observed that, in all concentrations and time points, the viability was above 95% (Figure 8B). Although some concentration had a significant difference compared with the control group, this difference cannot be considered biologically relevant. Because of the differentiation, the number of cells remained mostly equal in all time points (Figure 8A). The mechanism that the LIVE/DEAD employs to determine cytotoxicity differs from the colorimetric assays also performed in this paper. While colorimetric assays such as hemolysis, which quantifies the amount of hemoglobin solubilized after red blood cells lysis, or MTT, which quantifies the formazan produced by reduction of MTT by the oxidoreductases present in cells, the LIVE/DEAD assay can analyze the integrity of the cells by staining the cytoplasm if viable, or the nuclei if non-viable. High-throughput microscopy can transform a qualitative into a quantitative assay by analyzing the whole cell population.

## 3. Material and Methods

### 3.1. Materials

The α-amino-ω-hydroxy-terminated (NH_2_)-PEG-b-PCL used in this study was purchased from Polymer Source (Montreal, QC, Canada). The amphiphilic copolymer used had an average molar mass (Mn) of 5 × 10^3^ g/mol for the PEG block and 10.5 × 10^3^ g/mol for the PCL block (NH_2_–PEO–PCL), and an Mw/number average molecular weight (Mn) ratio of 1.50. Curcumin and L-DOPA were purchased from Sigma-Aldrich Co., LLC (St. Louis, MO, USA).

### 3.2. Preparation of NH_2_–PEO–PCL Nanoparticles

The NH_2_–PEO–PCL nanoparticle suspensions were prepared by using a nanoprecipitation and solvent displacement method, similar to that described by Numata et al., 2015 [24]. Following this method, NH_2_–PEO–PCL (10 mg) was dissolved in 0.5 mL of a mixture of THF/water 0.9/0.1 (*v*/*v*) and then stirred for 18 h. The polymer solution was then added in a drop-wise manner (5.0 mL/h) to ultrapure water, at room temperature (1 mL), under stirring (750 rpm). The remaining THF was rapidly eliminated by evaporation under reduced pressure at room temperature.

Curcumin and L-DOPA-loaded nanoparticles were prepared by cosolvent evaporation, as described above, except for dissolving curcumin and NH_2_–PEO–PCL in a mixture of THF/water 0.9/0.1 (*v*/*v*), and adding the solution in a drop-wise manner to a solution of 3 mg/mL of L-DOPA in 0.01 M monobasic potassium phosphate, adjusted to pH 3.0 with phosphoric acid 85%.

Curcumin-loaded nanoparticles were prepared by cosolvent evaporation, as described above, except for dissolving curcumin and NH_2_–PEO–PCL in a mixture of THF/water 0.9/0.1 (*v*/*v*). Then the nanoparticle suspension was centrifuged at 10,000 rpm for 5 min to remove the curcumin precipitate from the outside of the nanoparticles.

L-DOPA-loaded nanoparticles were prepared by cosolvent evaporation, as described above, except for adding the NH_2_–PEO–PCL solution in a drop-wise manner to a solution of 3 mg/mL of L-DOPA in 0.01 M monobasic potassium phosphate, adjusted to pH 3.0 with phosphoric acid 85%.

### 3.3. Coating of the NPs

The coating of NH_2_–PEO–PCL nanoparticle suspensions with glutathione (GSH) was prepared by using a method similar to Geldenhuys et al., 2015 [89]; 20 mg of glutathione was added to 1 mL of nanoparticle suspension in order to get a 2% *w*/*v* coating and allowed to sit at room temperature for at least 30 min before use in order to warrant a maximal GSH coating.

### 3.4. Characterization of NH_2_–PEO–PCL Nanoparticles

#### 3.4.1. Dynamic Light Scattering (DLS)

The size distribution, mean particle size, polydispersity index, and morphology of the nanoparticle suspensions in aqueous media were determined by using DLS at 25 °C by Static and Dynamic Light Scattering (SLS/DLS) with an ALV 5000 (ALV, Langen, Germany) providing a red helium–neon laser at a wavelength of 632.8 nm, utilizing 35 mW power. Then it was diluted in ultrapure water, and samples were placed in cylindrical measurement cells, which were immersed in a toluene bath at 25 °C. A very sensitive avalanche diode detected the scattered photons. In this study, the modulus of the scattering vector is denoted q and is equal to (4 pn/k)sin (h/2), where n represents the refractive index of pure water, h is the scattering angle, and k designates the light wavelength. Each experiment was performed during 120 s, and the scattered light was measured at different angles, ranging from 30° to 140°, with a 2.5° stepwise increase. The scattering intensity was corrected by considering the contributions of the solvent (water) and the toluene (standard), as well as the change of the scattering volume with the detection angle. The hydrodynamic radius (Rh) was determined by using the Stokes–Einstein equation, Rh = jBT/6 pgD, where jB is Boltzmann constant (in J/K), T is the temperature (in K), D is the diffusion coefficient, and g is the viscosity of the medium—pure water, in this case (g = 0.89 cP at 25 °C). Unloaded and loaded nanoparticle suspensions show no absorption at the wavelength used in light-scattering experiments. The data were acquired with the ALV correlator control software, and the distributions of the relaxation times, A(t), were obtained by using CONTIN analysis applied to the autocorrelation function, C(q, t).

#### 3.4.2. Nanoparticles Tracking Analysis (NTA)

Nanoparticle Tracking Analysis (NTA) experiments were performed by analyzing with a digital microscope LM10 System (NanoSight, Salisbury, UK). The samples were diluted in ultrapure water and then introduced into the chamber with a syringe. Each sample was illuminated with a 405 nm blue laser and separated. The video images of particles were analyzed by the NTA analytical software version 2.1 (NanoSight, Salisbury, UK). The particles were in movement under Brownian motion, and each video clip was captured over 60 s, at room temperature.

#### 3.4.3. Transmission Electron Microscopy

The morphology of the NH_2_–PEO–PCL nanoparticle was observed under a Philips CM200 microscope (Royal Philips, Amsterdam, Netherlands) operated at 120 kV. NH_2_–PEO–PCL nanoparticles in aqueous media were released on a glow-discharge carbon-coated copper grid, staining negative with 2% (*w*/*v*) uranyl acetate to dry completely.

#### 3.4.4. Zeta Potential Measurement

Nanoparticle samples were diluted in ultrapure water and placed in the electrophoretic cell, where a potential of ±150 mV was established. The laser-doppler anemometry with a Zetasizer Nano Series (Malvern Instruments, Worcestershire, UK) was used to determine the Zeta potential. The f potential values were used as the mean electrophoretic mobility calculated values, using Smoluchowski’s equation.

### 3.5. Stability of the GSH-Coated Nanoparticles with Added Salt

At different concentrations of NaCl solutions, the GSH-coated and uncoated nanoparticle suspensions were diluted. The effects of salt addition on the particle size and Zeta potential were monitored by using DLS, as described above.

### 3.6. Determination of Drug Loading and Entrapment Efficiency

The amount of L-DOPA loading and entrapment efficiency was estimated after the determination of the drug concentration in the nanoparticle suspensions by UV–HPLC, according to the method described by Pereira et al. (2012) [90], using a PerkinElmer Lambda 10 UV/Vis spectrophotometer (PerkinElmer, Inc., Waltham, MA, USA) at 280 nm. The loading efficiency (%) was estimated as the difference between the total concentration of L-DOPA found in the solution used for the nanoprecipitation. Then the drug concentration in the supernatant was obtained by the suspension ultrafiltration/centrifugation procedure (13,000 rpm for 15 min), using Amicon Centrifugal Filter Devices with Ultracel-3000 membrane (3 kDa, Millipore Corp., Burlington, MA, USA), after the nanoprecipitation to separate the free drug in the supernatant from the L-DOPA-loaded nanoparticles. The amount of encapsulated L-DOPA was calculated from the calibration curve established, using standard solutions of L-DOPA in the same solvent. The loading and entrapment efficiencies of L-DOPA were calculated according to the following formulas:(1)$$\begin{array}{c} Loading\\ Efficiency \end{array}\%=\frac{LD\mathrm{in}NPs-FreeLD}{TotalamountofLD}\times 100$$
(2)$$\begin{array}{c} Entrapment\\ Efficiency \end{array}\%=\frac{LD\mathrm{in}NPs-FreeLD}{TotalamountofNPs}\times 100$$
where *LD* = L-DOPA, and *NPs* = NH_2_–PEO–PCL nanoparticles.

Curcumin (CUR) loading and entrapment efficiency in the nanoparticle suspension were calculated after the drug concentration in the nanoparticle suspensions by UV–HPLC, according to the method described by Monton et al. (2016) [91], at 425 nm. The loading efficiency (%) was estimated as being the difference between the total amount of CUR added to the formulations and the total concentration of curcumin found in the nanoparticle suspensions after their complete dissolution in acetonitrile, deducted from the concentration of the drug in the supernatant, obtained by the suspension ultrafiltration/centrifugation procedure, using Amicon Centrifugal Filter Devices with Ultracel-100 membrane (100 kDa, Millipore Corp., Burlington, MA, USA), as described by Mazzarino et al. (2012) [37] and Scarano et al. (2015) [39]). The amount of CUR calculated from the calibration curve was established by using standard solutions of curcumin in the same solvent. The loading and entrapment efficiencies of CUR were calculated according to the following formulas:(3)$$\begin{array}{c} Loading\\ Efficiency \end{array}\%=\frac{CUR\mathrm{in}NPs-FreeCUR}{TotalamountofCur}\times 100$$
(4)$$\begin{array}{c} Entrapment\\ Efficiency \end{array}\%=\frac{CUR\mathrm{in}NPs-FreeCUR}{TotalamountofNPs}\times 100$$

*CUR* = curcumin, and *NPs* = NH_2_–PEO–PCL nanoparticles.

### 3.7. Freeze-Drying Stability

In order to minimize the physical changes of the NH_2_–PEO–PCL NPs, lactose, mannitol, sucrose, trehalose, and hydroxypropyl-β-cyclodextrin (HPbCD) were tested as cryoprotectants. Aqueous solutions were made at various concentrations of cryoprotectants and mixed with aliquots of the NH_2_–PEO–PCL NPs suspension before freeze-drying to obtain the final concentrations of cryoprotectant from 0.25 to 5%, *w*/*v*. The colloidal suspensions were placed inside a 1.5 mL microtube, frozen at −196 °C in liquid nitrogen, and then transferred immediately to a freeze dryer (Free Zone 6, Labconco, Kansas City, MO, USA) and lyophilized (−50 °C and 50 mbar) for 48 h. The freeze-dried cake was rehydrated by slowly injecting 1 mL ultrapure water onto the tube, stabilized for 5 min, and gently shaken to ensure complete disintegration and dissolution of the cake. The mean particle size of the original fresh nanoparticle was measured by DLS before (Si) and after (Sf) freeze-drying at 25 °C, and then it was compared and expressed as an Sf/Si % ratio.

### 3.8. Cytotoxicity Evaluations of the Loaded Nanoparticles

All cytotoxicity assays were performed by using drug concentrations of 1, 10, 25, 50, 75, and 100 µM.

#### 3.8.1. Erythrocyte Hemolysis Assay

To perform the erythrocyte hemolysis assay, 2 mL of peripheral blood was collected from two healthy donors into tubes containing EDTA anticoagulant. The erythrocytes were washed by separating them with centrifugation at 1500 G for 5 min, the supernatant was removed with micropipette, and the erythrocytes were suspended with saline, and this process was repeated twice. After the removal of the supernatant, 1 mL of the washed erythrocytes was added to 9 mL of saline, referred to now as the stock solution of erythrocytes. In 1.5 mL microtubes, 50 µL of the stock solution of erythrocytes was added and treated with 950 µL of saline containing the NH_2_–PEO–PCL NPs suspensions that were unloaded (UnNP) or loaded with curcumin and L-DOPA (LdCurNP) at a concentration of the drugs ranging from 1 to 100 µM (concentrations of the LdCurNPs), calculated regarding the drug L-DOPA. The negative control (established as 0% hemolysis) was treated with saline only, and the positive control (established as 100% hemolysis) was treated with distilled water only. The microtubes were homogenized and incubated in a shaker at 37 °C for 1 h. After the incubation time, the tubes were centrifuged 1500 G for 5 min, and the image of the tubes was recorded; the supernatant was collected and transferred to a 96-well plate and read at 405 nm wavelength in spectrophotometer. The hemolysis percentage was calculated according to the following formula:(5)$$Hemolysis\%=\frac{\mathrm{ABS}-\mathrm{ABSnc}}{\mathrm{ABSpc}-\mathrm{ABSnc}}\times 100$$
where *ABS* = the observed absorbance of the samples, *ABSnc* = the negative control absorbance, and *ABSpc* = the positive control absorbance.

#### 3.8.2. MTT Cytotoxicity Assay

Vero cells acquired from the Cell Bank of Rio de Janeiro (Duque de Caxias, RJ, Brazil) (BCRJ code: 0245/ATCC code: CCL-81) were cultivated with complete medium, DMEM-F12 supplemented with 10% fetal bovine serum, and 1% penicillin and streptomycin (100 U/mL and 100 µg/mL respectively), at 37 °C, with 5% CO_2_. They were seeded in 96-well plates (seeding density of 2 × 10^3^ cells per well). After 24 h, the medium was discarded, and the cells were treated with a complete medium containing the NH_2_–PEO–PCL NP suspensions that were either unloaded (UnNP) or loaded with L-DOPA and curcumin (LdCurNP), at a concentration ranging from 1 to 100 µM, for 24, 48, and 72 h. After the incubation time, 20 µL of 3-(4,5-Dimethyl-2-thiazolyl)-2,5-diphenyl-2H-tetrazolium bromide (MTT) solubilized in DMEM was added to each well (final concentration of MTT for each well was 0.5 mg/mL). The plates were incubated for 3 h at 37 °C; after this period, the medium was removed with a micropipette, and 100 µL of DMSO was added. The plates were placed onto a shaker for at least 30 min to solubilize the formazan crystals and finally read with 595 nm wavelength in a spectrophotometer. Data collected from three individual experiments, each with quadruplicates samples, were presented as a percentage of the observed absorbance of the control group.

PC-12 cells acquired from the European Collection of Authenticated Cell Cultures (ECACC) (Porton Down, Wiltshire, UK) (Catalogue No.: 88,022,401) were previously cultivated in suspension with RPMI 1640 supplemented with 10% fetal horse serum, 5% fetal bovine serum, and 1% penicillin and streptomycin (100 U/mL and 100 µg/mL respectively), at 37 °C, with 5% CO_2_. PC-12 cells can adhere to plastic pretreated with collagen type IV (Sigma Aldrich Catalogue Number C5533) (San Luis, MO, USA). The collagen coating was performed by adding 50 µL of a 0.01 mg/mL collagen solution in each well of a 96-well plate. Plates were left open in a laminar flow hood until total solvent evaporation. For sterilization, coated plates were exposed lidless to UV light for 1 h inside the laminar flow hood, sealed and maintained at 4 °C until use. PC-12 cells were seeded on a collagen type IV coated 96-well plate (seeding density of 2 × 10^3^ cells per well) with neuronal differentiation medium, RPMI 1640 supplemented with 1% of fetal horse serum, and 100 ng/mL of Nerve Growth Factor-7S (Sigma Aldrich catalog number N05013) (San Luis, MO, USA) for 17 days. Three-quarters of the differentiation medium was replaced twice a week. After neuronal differentiation, the medium was replaced by a differentiation medium containing the NH_2_–PEO–PCL NPs suspensions, followed by the same procedures described in this section.

#### 3.8.3. LIVE/DEAD Viability Assay

Vero cells, previously cultivated as described above, were seeded in 96-well plates (seeding density of 5 × 10^3^ per well). After 24 h, the medium was replaced, and the cells were treated with a complete medium containing the NH_2_–PEO–PCL NPs suspensions that were either unloaded (UnNP) or loaded with curcumin and L-DOPA (LdCurNP), at a concentration ranging from 1 to 100 µM, for 24, 48, and 72 h. After the incubation time, 20 µL of Hoechst 33,342, calcein AM, and Ethidium Homodimer-1 solubilized in DMEM-F12 were added to each well (fluorescence markers’ final concentration for each well was 2 µg/mL, 0.3 µM, and 0.6 µM, respectively). The plates were incubated for 30 min at 37 °C, and images were acquired with high-throughput microscopy (GE In Cell Analyzer 2000, Boston, MA, USA). The image acquisition protocol consisted of defined exposure time for each channel DAPI 650 ms, FITC 350 ms, and Texas Red 650 ms. These channels were used to observe the fluorescence of the Hoechst 33,342, calcein AM, and Ethidium Homodimer-1, respectively. In addition, each concentration having quadruplicate wells and images was acquired from four different fields in each well. Images were analyzed with an In Cell Analyzer Workstation v.3.7.3, using the cell nuclei for detection and the identification. Cell viability was determined by the positive staining with calcein, and cell death was marked with Ethidium Homodimer-1, respectively.

PC-12 cells were cultivated, seeded, and differentiated into neuron-like cells as described previously, followed by the same treatment and procedures described in this section.

## 4. Conclusions

In this study, NPs consisting of NH_2_–PEO–PCL were characterized as potential formulations to pass across the BBB for brain delivery of both L-DOPA and curcumin, transporting these active compounds simultaneously.

It is the first time that both L-DOPA and curcumin were presented together in the same nanoparticle that can act in treating Parkinson’s disease. L-DOPA is a dopamine precursor, and curcumin has antioxidant properties that could protect dopaminergic neurons.

The demonstrated results indicate that the developed biodegradable nanomicelles were blood compatible, with low cytotoxicity, and may be considered a promising novel therapy for treating Parkinson’s disease. Such a therapy represents a promising approach for future clinical applications and constitutes an exciting drug delivery system, with desirable features for brain delivery of a neurotransmitter precursor associated with an antioxidant molecule capable of producing a synergistic therapeutic effect. The advantages of these nanoparticles are that the treatment could be applied with lower concentrations, with the possibility of new routes of administrations, such as nasal, allied with higher treatment tolerance, due to the reduction of unwanted collateral effects.

However, to evaluate the anti-parkinsonian effects, future in vivo studies should be carried on to verify if these nanocarriers can reduce the dopaminergic neuron degeneration or the motor symptoms related to Parkinson’s disease.

## Figures and Tables

**Figure 1 molecules-27-02811-f001:**
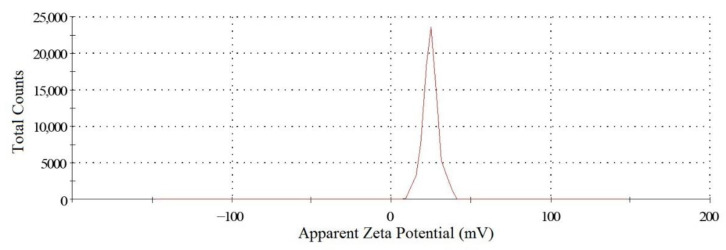
Surface zeta potential graph showing positive zeta potential value for NH_2_-PEO-PCL nanoparticle.

**Figure 2 molecules-27-02811-f002:**
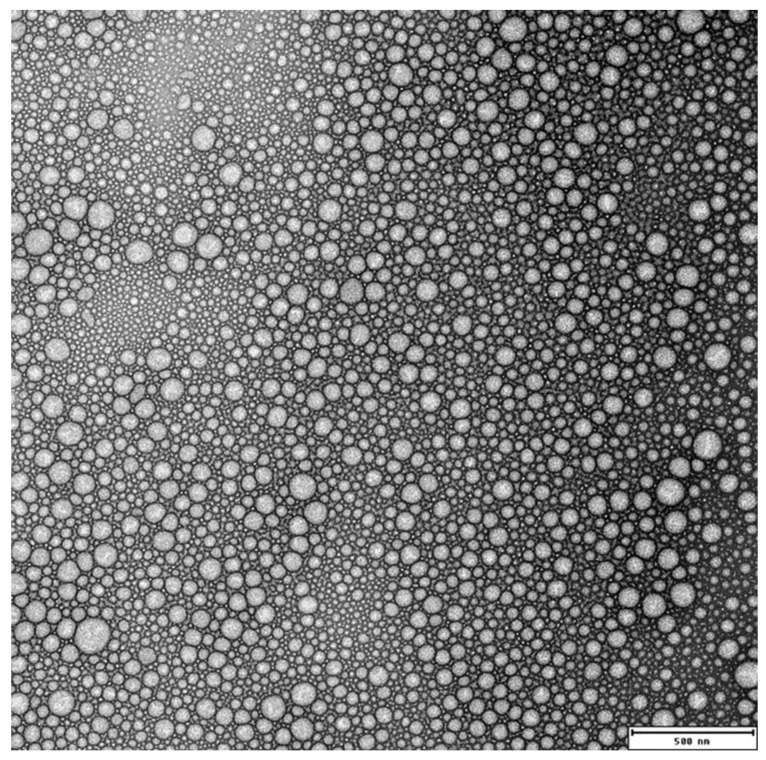
Transmission electron micrographs image of NH_2_-PEO-PCL nanoparticles (bar 500 nm).

**Figure 3 molecules-27-02811-f003:**
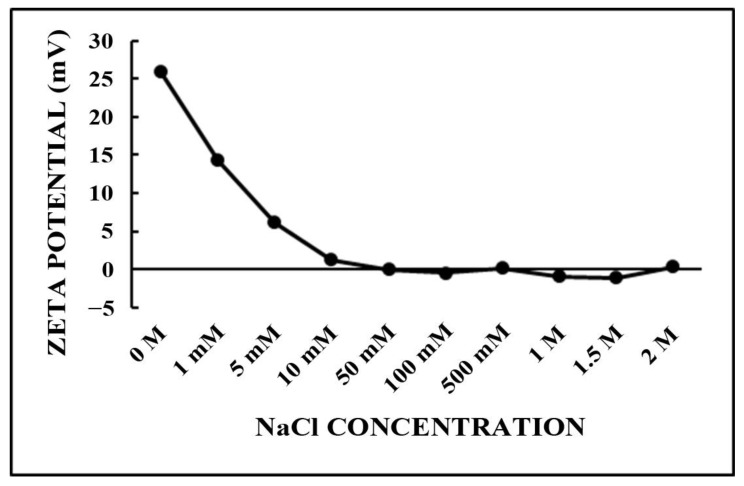
Zeta potential of NH_2_-PEO-PCL nanoparticles as a function of NaCl concentration.

**Figure 4 molecules-27-02811-f004:**
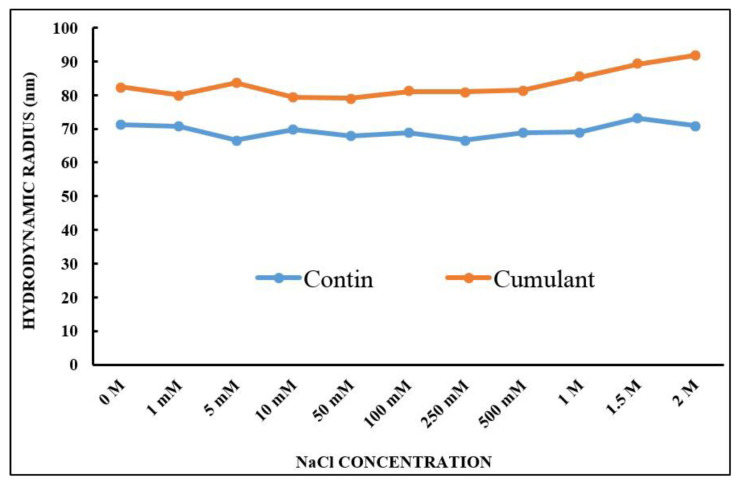
The hydrodynamic radius of NH_2_-PEO-PCL nanoparticles as a function of NaCl concentration.

**Figure 5 molecules-27-02811-f005:**
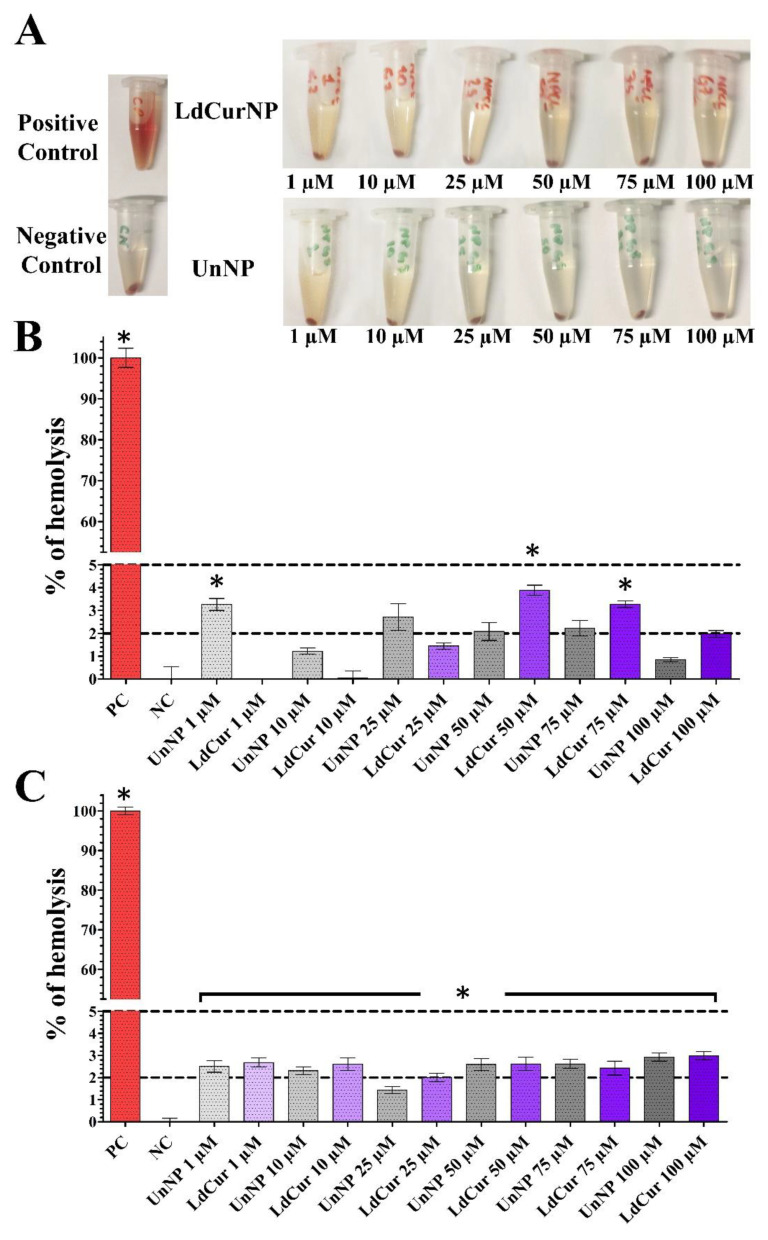
(**A**) Hemolysis assay microtubes after incubation with nanoparticles suspensions in different concentrations (1, 10, 25, 50, 75 and 100 µM) for 1 hour and centrifugation at 1500 G for 5 min. PC (Positive Control, distilled water), NC (Negative Control, saline), UnNPs (Unloaded Nanoparticles), LdCurNPs (L-DOPA and Curcumin- Loaded Nanoparticles). (**B**,**C**) The results presented are mean ± standard error of the hemolysis rate of samples 1 and 2, respectively. Dotted lines are the thresholds of 5% (slightly hemolytic) and 2% (non-hemolytic) hemolysis rate and * *p* < 0.05; n.s. *p* > 0.05 when compared with the negative control.

**Figure 6 molecules-27-02811-f006:**
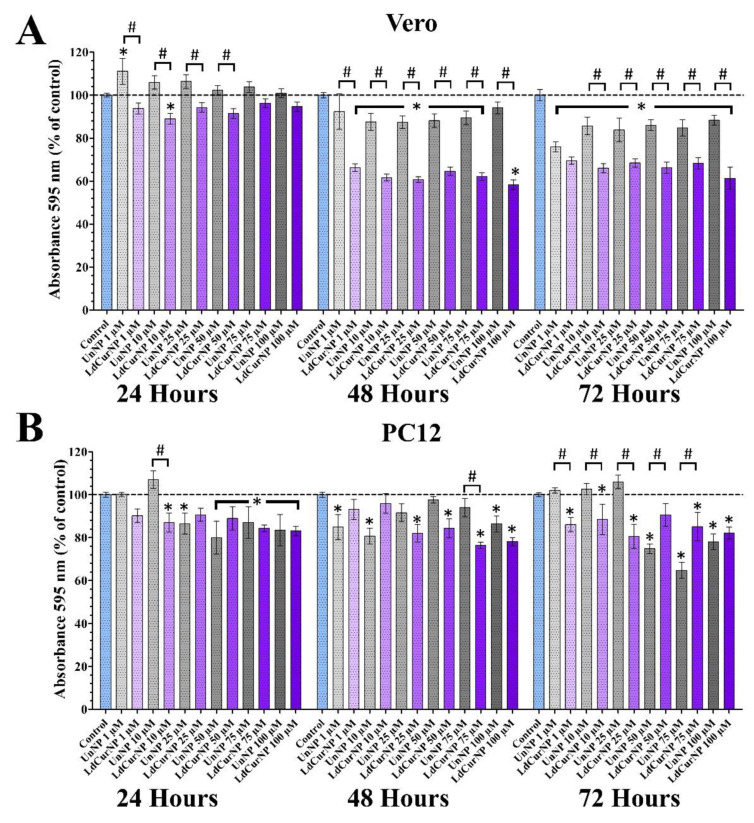
MTT assays of Vero and differentiated PC12 cells after 24, 48 and 72 h of incubation with the nanoparticles’ suspensions in different concentrations. Absorbance was determined in spectrophotometer at wavelength of 595 nm and values are presented as mean ± standard error of the % of control group (* and # *p* < 0.05; n.s. *p* > 0.05). (**A**) Vero cells treated with L-DOPA and Curcumin-loaded nanoparticles (LdCurNP) and unloaded nanoparticles (UnNP) suspensions in concentration of 1, 10, 25, 50, 75 and 100µM. (**B**) differentiated PC12 cells treated with LdCurNP and UnNP suspensions in concentration of 1, 10, 25, 50, 75 and 100 µM.

**Figure 7 molecules-27-02811-f007:**
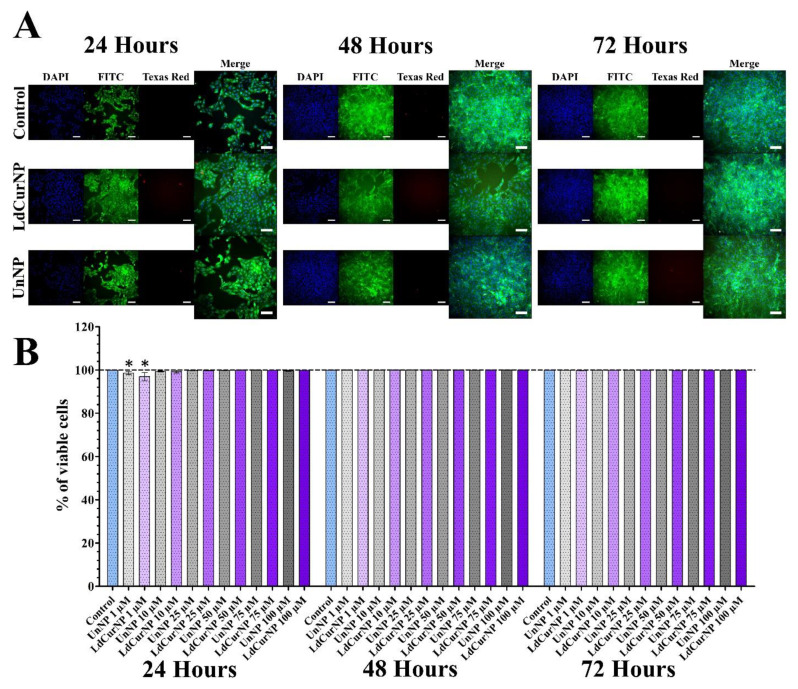
LIVE/DEAD viability assays of differentiated Vero cells after 24, 48 and 72 h of incubation with the nanoparticles’ suspensions in different concentrations. (**A**) Representative images of control, L-DOPA and Curcumin-loaded nanoparticles (LdCurNP) 100 µM and unloaded nanoparticles (UnNP) 100 µM. Cells nuclei were stained with Hoechst 33342 and observed with the DAPI channel, live cells were stained with calcein and observed in the FITC channel, dead cells were stained with ethidium homodimer-1 and observed in the Texas Red channel. (Scale bar 100 µm). (**B**) LdCurNP and UnNP suspensions in concentration of 1, 10, 25, 50, 75 and 100 µM, Values were presented as mean ± standard error of the % of viable cells (* *p* < 0.05; n.s. *p* > 0.05).

**Figure 8 molecules-27-02811-f008:**
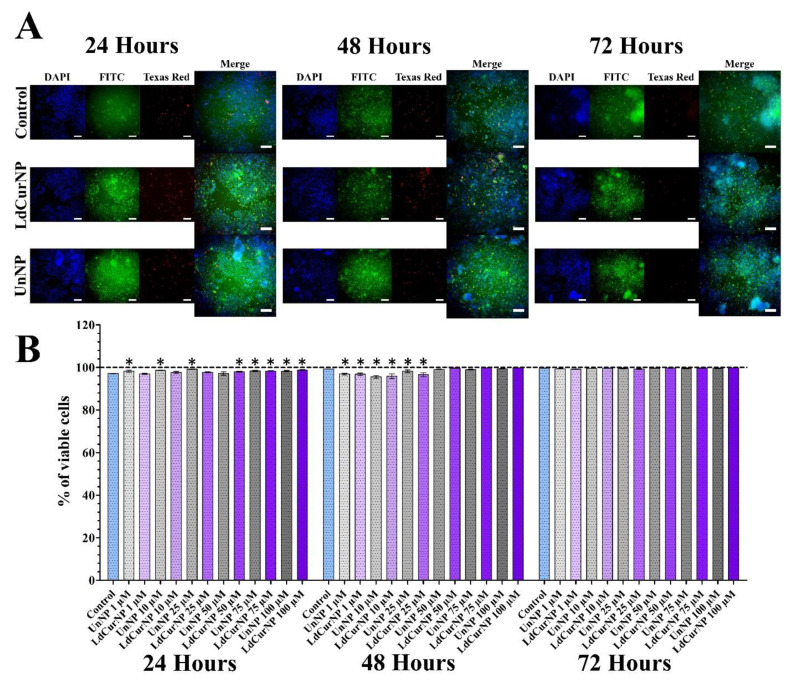
LIVE/DEAD viability assays of differentiated PC12 cells after 24, 48 and 72 h of incubation with the nanoparticles’ suspensions in different concentrations. (**A**) Representative images of control, L-DOPA, and Curcumin-loaded Nanoparticles (LdCurNP) 100 µM and Unloaded Nanoparticles (UnNP) 100 µM. Cells nuclei were stained with Hoechst 33342 and observed with the DAPI channel, live cells were stained with calcein and observed in the FITC channel, dead cells were stained with ethidium homodimer-1 and observed in the Texas Red channel. (Scale bar 100 µm). (**B**) LdCurNP and UnNP suspensions in concentration of 1, 10, 25, 50, 75 and 100 µM, Values were presented as mean ± standard error of the % of viable cells (* *p* < 0.05; n.s. *p* > 0.05).

**Table 1 molecules-27-02811-t001:** Physicochemical parameters of NH_2_–PEO–PCL, GSH adsorbed and L-DOPA and curcumin-loaded nanoparticles in aqueous solution.

Sample	2 Rh (nm)	NTA (nm)	PDI	Zeta Potential (mV)
NH_2_–PEO–PCL (UnNP)	1.17 × 10^2^ ± 8.4	9.95 × 10 ± 7.3	0.22	+25.6 ± 0.45
GSH NH_2_–PEO–PCL (UnNP)	1.28 × 10^2^ ± 2.7	1.05 × 10^2^ ± 1.8	0.21	+10.4 ± 0.73
L-DOPA + CUR NH_2_–PEO–PCL (LdCurNP)	1.33 × 10^2^ ± 6.4	1.23 × 10^2^ ± 4.0	0.24	+24.6 ± 0.6
GSH L-DOPA + CUR NH_2_–PEO–PCL (LdCurNP)	1.45 × 10^2^ ± 3.2	1.34 × 10^2^ ± 5.0	0.30	+6.4 ± 0.53

**Table 2 molecules-27-02811-t002:** Drug loading and encapsulation efficiency of NH_2_–PEO–PCL NPs loaded with curcumin, L-DOPA, or both.

Nanoparticle	Drug Loading	Encapsulation Efficiency
Curcumin-loaded NP	98.3% ± 0.9%	19.8% ± 0.2%
L-DOPA-loaded NP	12% ± 1.4%	3.6% ± 0.4%
L-DOPA and Curcumin-loaded NP (LdCurNP)	10.4 ± 1.5% (of L-DOPA)	3.1 ± 0.5% (of L-DOPA)
L-DOPA and Curcumin-loaded NP (LdCurNP)	97.7 ± 1.0% (of Curcumin)	19.5 ± 0.2% (of Curcumin)

**Table 3 molecules-27-02811-t003:** Hemolysis assays of the L-DOPA and curcumin-loaded nanoparticles (LdCurNPs) and unloaded nanoparticles (UnNPs) results presented as mean ± standard error.

	% Hemolysis (LdCurNP)		% Hemolysis (UnNP)	
µM	Sample 1	Sample 2	Sample 1	Sample 2
1	−0.16 ± 0.11	2.68 ± 0.21	3.27 ± 0.26	2.50 ± 0.26
10	0.04 ± 0.32	2.61 ± 0.28	1.22 ± 0.13	2.31 ± 0.18
25	1.45 ± 0.14	2.00 ± 0.19	2.71 ± 0.58	1.43 ± 0.16
50	3.89 ± 0.22	2.62 ± 0.30	2.08 ± 0.39	2.59 ± 0.26
75	3.28 ± 0.14	2.43 ± 0.32	2.23 ± 0.34	2.62 ± 0.21
100	1.97 ± 0.15	2.98 ± 0.19	0.85 ± 0.09	2.93 ± 0.18

## Data Availability

The data presented in this study are available in Appendix A.

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
