# Peer review of "Biodegradable Nanoparticles Loaded with Levodopa and Curcumin for Treatment of Parkinson’s Disease"

_molecules, 2022, doi:10.3390/molecules27092811_

Round 1

Reviewer 1 Report

The manuscript presented by B.F. Mogharbel et al. contains  interesting and valuable results which deserve to be published. However the paper have to be reconsidered as despite the fact the the introduction and the methods chapters are properly presented the results chapter is very hard to read and to understand. Yes, the results have to be compared with those from the literature, but as it is now it is mostly a literature review from which it is difficult to see the original results. In addition for me was difficult to see in the figures what the authors claims and moreover what means A, B and C in the figures. In one moment there is reference to table 1, but table 1 is missing.

The Conclusion Chapter should be developed and highlighting the novelty and the originality of the results.

Overall the article should be carefully read by the authors and to correct some minor errors (e.g. subchapter numbers from chapter 2 and English errors) and at the same time to clearly present the original results (chapter 3) and the conclusions. 

Author Response

REVIEWER 1

 Dear Reviewer,

  1. The manuscript presented by B.F. Mogharbel et al. contains interesting and valuable results which deserve to be published. However, the paper has to be reconsidered as despite the fact the introduction and the methods chapters are properly presented the results chapter is very hard to read and to understand. Yes, the results have to be compared with those from the literature, but as it is now it is mostly a literature review from which it is difficult to see the original results.

Answer: We agree with you, we have briefly explained other studies, and we emphasize our results, as you can see in the main text. 

  1. In addition for me was difficult to see in the figures what the authors claims and moreover what means A, B and C in the figures.

Answer: We apologize for the figures' information on A, B, and C. We did our best for better legends. 

  1. ..one moment, there is reference to table 1, but table 1 is missing.

Answer: We apologize about missing tables 1and 2, and they were included.

4.The Conclusion Chapter should be developed and highlighting the novelty and the originality of the results.

Overall, the article should be carefully read by the authors and to correct some minor errors (e.g. subchapter numbers from chapter 2 and English errors) and at the same time to clearly present the original results (chapter 3) and the conclusions. 

Answer: We are grateful for your carefully read; we have made the corrections the emphasize our results.

Answer: We did English corrections. We want to inform you that one of the authors was Canadian and has the English domain. She revised the English carefully.

We sincerely thank you for your constructive criticisms and valuable comments, which helped revise the manuscript.

Corresponding author,

Reviewer 2 Report

The authors have performed a good work and it can be accepted after some English corrections

Author Response

REVIEWER 2

Dear Reviewer,

  1. The authors have performed a good work and it can be accepted after some English corrections

Answer: we are thankful for your carefully read and attention to our manuscript. Also, we did English corrections. We would like to inform you that one of the authors was Canadian and has the English domain. She revised the English carefully.

We sincerely thank you for your constructive criticisms and valuable comments, which helped revise the manuscript.

Corresponding author,

Reviewer 3 Report

The present work reports the synthesis of a positively charged NH2-PEO-PCL diblock copolymer modified with glutathione (GSH) on the outer surface, leading in a synergistic delivery of LDOPA curcumin capable of crossing the blood-brain barrier.

Introduction:

1- "The blood-brain barrier (BBB) is a structure that separates the central nervous systemfrom the circulatory system restricting the diffusion of molecules into the brain while facilitating the transport of essential nutrients across endothelial cells. This BBB provides aneffective defense system against potentially harmful chemicals, toxins, and infections that may enter the brain. However, it is a disadvantage when attempting to treat brain disorders once this pathway limits the access of many therapeutics and macro molecular structures"

2-Curcumin (CUR) is a natural low molecular weight hydrophobic natural polyphenolic phytoconstituent and the perennial herb Curcuma longa L. which was first isolated from the rhizome of turmeric in 1815 and then synthesized by Lampe in 1910[11]. It was described pharmacological activities, including antioxidant, anti-inflammatory, antimicrobial, anti-amyloid, antitumor properties, anti-apoptosis, and no signs of toxicity[12].
However, the clinical application of CUR is restricted due to its poor solubility in aqueous solution, poor oral bioavailability, and extensive first-pass metabolism. In recent years, the development of nanotechnology-based delivery systems has overcome these pharmaceutical issues related to the delivery of CUR in biological systems and demonstrated its neuroprotective effect in Parkinson’s disease models[

This is a research paper will be published in a prominent journal, and such elementary material is only necessary for students' text books; thus, delete it and replace it with succinct, scientific, and specialized information.

Methods:

Take care of subscripts and superscripts. NPs: NH2-PEO-PCL nanoparticles 

Results and Discussion

Whole section is full of supporting references and 80% writing is about other reported works, If I am not mistaken its an original research article and not the review article so why authors have no focus on their own performed work?

Supporting references are good but not everywhere and in every aspect. Start all parts of results discussion with your own results and at end try to compare with related reported work.

This section need extensive improvement.

Author Response

REVIEWER 3

Dear Reviewer,

1.The present work reports the synthesis of a positively charged NH2-PEO-PCL diblock copolymer modified with glutathione (GSH) on the outer surface, leading in a synergistic delivery of LDOPA curcumin capable of crossing the blood-brain barrier.

Introduction:

1- "The blood-brain barrier (BBB) is a structure that separates the central nervous system from the circulatory system restricting the diffusion of molecules into the brain while facilitating the transport of essential nutrients across endothelial cells. This BBB provides an effective defense system against potentially harmful chemicals, toxins, and infections that may enter the brain. However, it is a disadvantage when attempting to treat brain disorders once this pathway limits the access of many therapeutics and macro molecular structures"

2-Curcumin (CUR) is a natural low molecular weight hydrophobic natural polyphenolic phytoconstituent and the perennial herb Curcuma longa L. which was first isolated from the rhizome of turmeric in 1815 and then synthesized by Lampe in 1910[11]. It was described pharmacological activities, including antioxidant, anti-inflammatory, antimicrobial, anti-amyloid, antitumor properties, anti-apoptosis, and no signs of toxicity [12].
However, the clinical application of CUR is restricted due to its poor solubility in aqueous solution, poor oral bioavailability, and extensive first-pass metabolism. In recent years, the development of nanotechnology-based delivery systems has overcome these pharmaceutical issues related to the delivery of CUR in biological systems and demonstrated its neuroprotective effect in Parkinson’s disease models [

This is a research paper will be published in a prominent journal, and such elementary material is only necessary for students' textbooks; thus, delete it and replace it with succinct, scientific, and specialized information.

Answer: We agree with you and attended your suggestion about excluding much information we believe the readers know. See in the main text. 

2.Methods:

Take care of subscripts and superscripts. NPs: NH2-PEO-PCL nanoparticles 

Answer: Thank you for this essential observation. We did all corrections.

3.Results and Discussion

Whole section is full of supporting references and 80% writing is about other reported works, If I am not mistaken it’s an original research article and not the review article so why authors have no focus on their own performed work?

Answer: Thank you for this observation. You can see it in the main text. We reduced the text as well as the references.

4.Supporting references are good but not everywhere and in every aspect. Start all parts of results discussion with your own results and at end try to compare with related reported work. This section needs extensive improvement.

Answer: Thank you for this observation. You can see it in the main text. We reduced the text as well the references: 109 to 91. 

We sincerely thank you for your constructive criticisms and valuable comments, which helped revise the manuscript.

Corresponding author,

Reviewer 4 Report

Dear Authors,

This is an excellent manuscript, of very good experimental design, precise and clear explanations of all parts of the text, clear conclusion. I can only wish that your patent will soon be found in clinical trials.

I would only recommend that you consider the first sentence on page 2 "It can be caused by dementia or depression [1]".
Parkinson's disease is associated or linked but not directly caused by dementia or depression.

All the best!

Author Response

REVIEWER 4

Dear Reviewer,

1.This is an excellent manuscript, of very good experimental design, precise and clear explanations of all parts of the text, clear conclusion. I can only wish that your patent will soon be found in clinical trials.

Answer: We are so glad to this comment, why it was scientific stimulation. Thanks a lot.

2.I would only recommend that you consider the first sentence on page 2 "It can be caused by dementia or depression [1]".
Parkinson's disease is associated or linked but not directly caused by dementia or depression.

Answer: We agree with you; it was a fine observation. We made this correct approach with Parkinson's and dementia or depression in the main text.

3.All the best!

Also, we did English corrections. We would like to inform you that one of the authors was Canadian and has the English domain. She revised the English carefully.

Answer: Thanks a lot.

We sincerely thank you for your constructive criticisms and valuable comments, which helped revise the manuscript.

Corresponding author,

Round 2

Reviewer 3 Report

Authors have improved the manuscript.